# Differential plasma exosomal long non-coding RNAs expression profiles and their emerging role in E-cigarette users, cigarette, waterpipe, and dual smokers

Gagandeep Kaur[1], Kameshwar Singh[1], Krishna P. Maremanda[1], Dongmei Li[2], Hitendra S. Chand[3], Irfan Rahman[1]*

**1** Department of Environmental Medicine, University of Rochester Medical Center, Rochester, NY, United States of America, **2** Department of Clinical & Translational Research, University of Rochester Medical Center, Rochester, NY, United States of America, **3** Department of Immunology and Nanomedicine, Florida International University, Miami, FL, United States of America

\* irfan_rahman@urmc.rochester.edu

**Data Availability Statement:** All authors confirm the availability of data and materials online/free access to readers as included in this manuscript.

## Abstract

Long non-coding RNAs (lncRNAs) are the varied set of transcripts that play a critical role in biological processes like gene regulation, transcription, post-transcriptional modification, and chromatin remodeling. Recent studies have reported the presence of lncRNAs in the exosomes that are involved in regulating cell-to-cell communication in lung pathologies including lung cancer, chronic obstructive pulmonary disease (COPD), asthma, and idiopathic pulmonary fibrosis (IPF). In this study, we compared the lncRNA profiles in the plasma-derived exosomes amongst non-smokers (NS), cigarette smokers (CS), E-cig users (E-cig), waterpipe smokers (WP) and dual smokers (CSWP) using GeneChip™ WT Pico kit for transcriptional profiling. We found alterations in a distinct set of lncRNAs among subjects exposed to E-cig vapor, cigarette smoke, waterpipe smoke and dual smoke with some overlaps. Gene enrichment analyses of the differentially expressed lncRNAs demonstrated enrichment in the lncRNAs involved in crucial biological processes including steroid metabolism, cell differentiation and proliferation. Thus, the characterized lncRNA profiles of the plasma-derived exosomes from smokers, vapers, waterpipe users, and dual smokers will help identify the biomarkers relevant to chronic lung diseases such as COPD, asthma or IPF.

## Introduction

A large proportion of human genome encodes for the long non-coding RNAs (lncRNAs), a diverse set of transcripts that do not encode for any functional protein. LncRNA transcripts are larger than 200 nucleotides in length and include, intronic lncRNA, long intergenic non-coding RNAs (lincRNAs), pseudogenic transcripts, circular RNAs (circRNA), long enhancer noncoding RNAs (eRNAs), natural antisense transcripts (NATs), and transcribed ultra-conserved regions (TUCRs) [1, 2]. Studies have shown that lncRNAs play crucial role in various

The microarray data and/or analyzed during the current study are available from Gene Expression Omnibus accession number GSE160769 (https://www.ncbi.nlm.nih.gov/geo/query/acc.cgi?acc=gse160769).

**Funding:** This work was supported in part by the National Institutes of Health (NIH) (NIH 1R01HL135613), the National Cancer Institute of the NIH and the Food and Drug Administration (FDA) Center for Tobacco Products (U54CA228110). HSC was also supported by NIH (R21AI144374). The funders had no role in study design, data collection and analysis, decision to publish, or preparation of the manuscript.

**Competing interests:** The authors have declared that no competing interests exist.

**Abbreviations:** E-cigs, E-cigarettes; COPD, Chronic obstructive pulmonary disease; IPF, Idiopathic Pulmonary Fibrosis; ILD, Interstitial Lung Disease; NS, Normal/non-smokers; CS, Cigarette smokers; WP, Waterpipe; CSWP, Dual (Cigarette and waterpipe) smokers; lncRNAs, Long non-coding RNAs; miRNA, micro RNA.

biological processes including gene regulation, gene transcription, post-transcriptional modification, and chromatin remodeling in health and disease [3, 4]. Current lncRNA database report that there are around 127,802 putative lncRNAs encoded by human genome and are highly abundant than the number of protein-coding genes [5], however, not much is known about their role in regulating the physiological responses to environmental exposures such as tobacco smoke.

In the last few years, lncRNAs are identified in the exosomes (the extracellular vesicles known to be essential for cell-to-cell communication) [6, 7]. Exosomes are the smallest subtype of extracellular vesicles (30–150 nm in size) surrounded by phospholipid bilayer [7, 8]. They are present in various biofluids including blood plasma, breast milk, urine, saliva, and bronchoalveolar lavage [9, 10]. Importantly, exosomal lncRNAs are being extensively investigated as potential diagnostic biomarkers for various disease conditions, importantly cancers [7, 10]. Yet, despite the growing interest in this area, little is known about their physiological function and role in other inflammatory disease conditions.

In this respect, evidences of involvement of exosomal lncRNA in lung pathologies including lung cancer, tuberculosis (TB), chronic obstructive pulmonary disease (COPD), asthma, and interstitial lung disease (ILD) are increasing [1, 11–15]. It has been demonstrated that the exosomal microRNAs (miRNA) and lncRNA profiles among patients with pulmonary pathologies are quite distinct than healthy individuals [12, 14, 16, 17]. Microarray analyses of lncRNA population in the lung tissues from non-smokers without COPD and smokers with or without COPD detected alteration in distinct classes of lncRNA., cigarette smoke exposure is associated with changes in lncRNAs which play an important role in the activation of metabolic pathways, whereas COPD onset and progression was attributed to activation of lncRNAs involved in intermediary metabolism and immune system processes [18]. It is likely that tobacco smoke exposure can alter the lncRNA profile of the lung tissue that can be involved in the pathogenesis of lung diseases.

Smoking habits amongst adults and adolescents are not only confined to combustible tobacco cigarettes but now has expanded to other tobacco products. In the last few decades, the popularity of E-cigarette and related products is on a rise amongst young adults in the US [19, 20]. At the same time, the prevalence of 'hookah' or waterpipe is also increasing [21, 22]. Considering this and based on our recent observations of differential regulation of miRNA [23], we compared the expression profiles of plasma-derived exosomal lncRNAs from non-smokers (NS), cigarette smokers (CS), E-cig users (E-cig), waterpipe (WP) users and dual (cigarette and waterpipe) smokers (CSWP). The objective of this study was to characterize the lncRNA content of the plasma-derived exosomes from a representative population of these tobacco product users to help identify novel biomarkers, which may be relevant to the associated pulmonary pathologies, such as COPD, asthma, and IPF.

## Materials and methods

### Ethics approval and consent to participate

The Institutional Review Board (IRB) /Research Subject Review Board (RSRB) committee at the University of Rochester Medical Center, Rochester, NY with an approval number CR00002635, approved all the protocols and procedures. All the study participants recruited for the study signed an informed written consent form before recruitment and sample collection (August 2017 until February 2020).

Human study protocol: Yes; Animal study protocol: None; Institutional biosafety approvals: Yes. The University of Rochester Institutional Biosafety Committee approved the study (study approval number: Rahman/102054/09-167/07-186; identification code: 07–186; date of approval: 5 January 2019).

## Study design and subject recruitment

We employed the blood samples collected during the cross-sectional pilot study from 2017–2019 for exosome isolation and lncRNA expression analyses. A detailed description of the study subject recruitment and their inclusion/eligibility and exclusion criteria were reported previously [24, 25]. In brief, n = 6–8 subjects (equal number of males and females based on local demography) were selected in each group. Our subject groups included; (a)non-smokers/non-users, (b) E-cigarette users, (c) cigarette smokers, (d) waterpipe smokers, and (e) dual smokers comprising both waterpipe and cigarette smokers. The study participants were recruited at the General Clinical Research Center at the University of Rochester Medical Center, Rochester, NY with the help of local newspaper and magazine advertisements. Each subject filled a questionnaire containing information about demographics, clinical symptoms, E-cig use, and vaping history [25, 26]. We collected 20–25 ml blood by venipuncture from smokers (CS), vapers (E-cig), waterpipe (WP) users, and subjects indulged in both cigarette smoking and waterpipe use (CSWP) as previously shown [23, 26]. Subjects included for the study were between 18–65 yrs. in age and reported at least 6 months of E-cig or tobacco use (at least once daily). Subjects with pre-existing conditions (bronchiectasis, lung cancer, COPD, asthma, cystic fibrosis, congestive heart failure, or other chronic illness) or using any medication (systemic corticosteroid or anti-inflammatory therapy) were not included in the study. Likewise, pregnant or lactating females were not recruited in this study. The smoking status of the subjects was confirmed by performing plasma cotinine assay [24, 25].

## Plasma exosome isolation

We employed commercially available kit from Norgen Biotek (Cat# 57400; Ontario, Canada) to isolate exosomes from human plasma. Plasma exosomes were isolated as per manufacturer's protocols as described earlier [23, 27]. In brief, 1 ml plasma was centrifuged at 400 g (~2000 rpm) for 2 min. The supernatant was collected and mixed with 3 ml nuclease-free water,100 μl ExoC buffer and 200 μl of Slurry E and incubated for 5 minutes at room temperature. Thereafter, the tubes were centrifuged at 2000 rpm for 2 minutes and the supernatant was discarded. Next, 200 μl of ExoR buffer was added to the slurry pellet and mixed. This solution was incubated at room temperature for 5 minutes after which the tubes were again centrifuged at 500 rpm for 2 minutes. Finally, the supernatant was transferred to a mini filter spin column to elute the exosomal fraction following centrifugation at 6000 rpm for 1 minute. The eluted exosomes were stored at -80˚C until further use.

## Exosome characterization

Transmission electron microscopy (TEM) was used to visualize the isolated exosomes and nanoparticle tracking analysis was performed to analyze the particle size and concentration as described [23].

## Exosomal RNA extraction

Total RNA from exosomes was isolated using Exosomal RNA isolation kit (Norgen Bioteck Corporation, Cat# 58000) as per the manufacturer's protocol. The detailed procedure has been published earlier [23]. The RNA quality and quantity were checked using NanoDrop ND-1000 spectrophotometer (Thermo Fisher Scientific, Wilmington, DE).

## Transcriptome library preparation

We employed the GeneChip™ WT Pico kit (Cat# 902622 and 902623, Applied Biosystems, Foster city, CA) for transcriptome profiling of the RNA isolated from various experimental

groups. This highly sensitive and flexible assay is used to assess the RNA expression from as little as 100 picogram of total RNA. The assay ensures strand specificity and can perform multiple layers of analysis in combination with transcriptome arrays to accurately measure gene- or transcript-level expression of coding as well as long non-coding RNA [28].

The Microarray & NextGen Sequencing Core at the Center for Functional Genomics at SUNY, Albany performed the library preparation and microarray hybridization. In brief, 100 pg of high-quality total RNA was reverse transcribed and amplified per manufacturer's protocol. RNA amplification was achieved using low-cycle PCR followed by linear amplification with the help of T7 in vitro transcription (IVT) technology. The obtained complimentary RNA was then converted to biotinylated sense-strand cDNA targets for hybridization and unbiased coverage of the transcriptome.

## Microarray and computational analysis

The single-stranded cDNA obtained from the previous step was mixed with the hybridization mixture and loaded onto the GeneChip Cartridge array provided with the kit and incubated for 16 hrs. at 45˚C and 60 rpm rotation in GeneChip™ Hybridization Oven. Thereafter, the array was washed, stained and scanned as per the manufacturer's protocol. The data files obtained after the scan were analyzed using the Transcriptome Analysis Console (TAC 4.0.1) Software using the Data flow worksheet as provided by Affymetrix.

## Statistical analysis

The long non-coding RNA (lncRNA) data from two batches were first normalized using the *rma* function in the R/Bioconductor. The annotation of the lncRNAs were obtained from the Clariom D human array from Affymetrix through the *annotateEset* function in R/Bioconductor. Boxplot was used to examine the distribution of the normalized lncRNA data from all samples. Multidimensional scaling (MDS) plot was used to determine the spatial distance between the lncRNA samples. Linear regression models with empirical Bayes approach was then used to fit the normalized lncRNA data with the batch effects and initial RNA quantity differences were adjusted. Linear contrasts within the linear regression model framework was further used to examine the differences in the comparisons between different groups of interest. Volcano plots were generated to highlight significant lncRNAs with significant fold changes (> 2-fold changes) and raw P-values (less than 0.0001) using the *EnhancedVolcano* function in R/Bioconductor. Heatmaps were generated to show the lncRNA expressions in different groups using the *pheatmap* function with *ward. D2* method for clustering the significant lncRNAs in R/Bioconductor. To examine the overlaps in identified lncRNAs from related group comparisons, *venn.diagram* function in R/Bioconductor was used.

## Results

### lncRNAs expression profiling in plasma exosomes from non-smokers, cigarette smokers, E-cig users, waterpipe users and dual smokers

To understand the role of exosomal lncRNA in regulating downstream signaling amongst smokers, vapers, waterpipe and dual users, we first isolated the plasma exosomes from the blood of six study participants from each group. The isolated exosomes were characterized using immunoblotting and transmission electron microscopy (TEM) as published earlier [23]. Next, total RNA was isolated from the exosomal fractions and the expression profiling of the lncRNAs was performed.

**Box-Plot of Normalized Expression Values**

**Fig 1. Validation of differential expression of plasma exosomal lncRNAs amongst non-smokers, smokers, E-cig users, waterpipe users and dual smokers.** Box-plots (normalized expression values using log2) were used to evaluate the variation in the expression and overall characteristic distribution of lncRNAs in plasma exosomes from non-smokers, cigarette smokers, electronic cigarette users, waterpipe users and dual smokers. Box plot showing the distribution of maximum, minimum and percentile values for lncRNA expression amongst each sample. NS, non-smokers; CS, cigarette smokers; E-cig, Electronic cigarette users; WP, waterpipe smokers; CSWP, dual smokers, i.e., cigarette and waterpipe smokers.

The raw expression data from the lncRNAs expression profiling was normalized and plotted as a box plot as shown in **Fig 1**. We did not observe any changes in the data distribution across the groups after normalization and proceeded further with the analyses.

## Multidimensional scaling (MDA) plot reveals the batch variations in the lncRNA expression data amongst the experimental groups

To visualize the similarities between each sample within the groups, we employed multidimensional scaling (MDS). MDS plot revealed the batch variations within the samples. The dual smokers (CSWP) group had minimum batch effects in the lncRNA expression as seen by their close clustering in the MDS plot based on batch 2 analyses. Rest of the four groups–NS, CS, E-cig and WP- showed variations (due to batch effect) as seen in **Fig 2**.

## lncRNA from plasma exosomes of non-smokers, smokers, E-cig users, waterpipe users and dual smokers exhibit distinct expression profiles

We generated volcano plots showing pairwise comparisons of the differential expression profiles of the lncRNAs from various experimental groups (**Fig 3**). The volcano plots were plotted such that each of the significant differentially expressed lncRNAs in the treatment group versus the control are denoted as colored dots. Any fold change greater than ±1 on the logarithmic (base2) scale was considered significant in our study, and is denoted by green dots on the graph. A threshold of 4 for raw p-value on the logarithmic scale (base 10) was considered to be significant and has been plotted as blue and red dots in the graph. Importantly, the red dots denote the lncRNAs that are differentially expressed with a fold change greater than ±1 as compared to control and are statistically significant with raw p-value p<0.0001.

**MDS Plot**

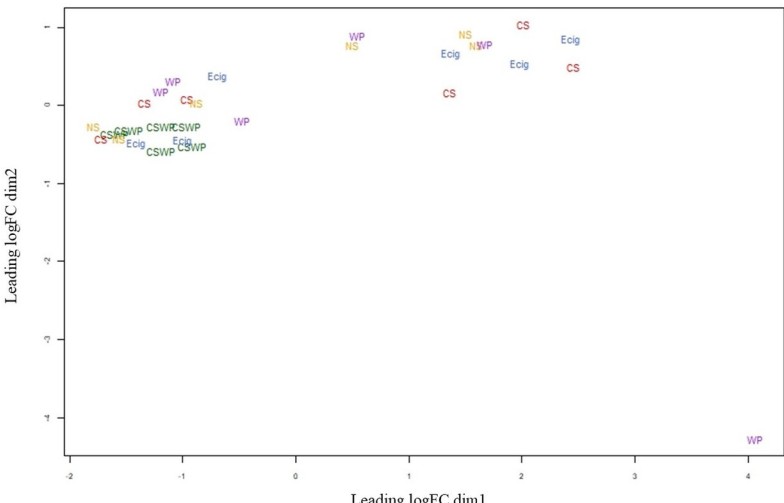

**Fig 2. Multidimensional scaling plot (MDS) for the study group shows batch variations.** MDS plot based on differential lncRNA expression in individual samples of non-smokers (NS), cigarette smokers (CS), E-cigarette users (E-cig), waterpipe smokers (WP) and dual smokers (CSWP).

## Hierarchical clustering shows significant variations in the lncRNA expression in plasma exosomes from non-smokers, smokers, E-cig users, waterpipe users and dual smokers

Next, we plotted heatmaps showing the differential expression of lncRNAs comparing CS vs. NS, E-cig vs. NS, WP vs. NS, CSWP vs. NS, CS vs. E-cig, WP vs. E-cig, CSWP vs. E-cig, CS vs. WP, CSWP vs. CS and CSWP vs. WP, respectively in **Fig 4A–4J**. Each row in the heat map represents individual lncRNAs and each column denotes individual sample. The color scale indicates the relative expression level of lncRNAs with green showing upregulation while red representing downregulation of gene expression. A detailed information about the identified differentially expressed genes along with fold change and p-value information is provided in **Table 1**. Briefly, the following observations were made from various pairwise comparisons:

**Smokers vs non-smokers.** We found 7 differentially expressed lncRNAs in the exosomes from blood plasma from CS as compared to NS. Two of those lncRNAs were upregulated while the rest where downregulated amongst smokers as compared to the NS controls (**Fig 4A**).

**E-cig users vs non-smokers.** In total, we observed 13 differentially expressed lncRNAs-eight upregulated and five downregulated- on comparing E-cig users with NS (**Fig 4B**). Interestingly, we found approximately 4-fold increase in the expression of BNIP3L, Bcl2 interacting protein 3-like protein (Fold change = 3.74; p = 2.8E-05) amongst the E-cig users as compared to the non-smoking controls. We further found an increase in the expression of lncRNA belonging to members of RNA binding [lozorby (Fold change = 1.64; p = 2.96E-05)] and endonuclease reverse transcriptase [zoyberby (Fold change = 1.81; p = 4.40E-05) and flarchoy (Fold change = 1.26; p = 7.32E-05)] protein family.

**Waterpipe user vs non-smokers.** We identified 18 differentially expressed lncRNAs in plasma exosomes on comparing WP group with NS (**Fig 4C**). Of these, 9 were upregulated while 9 others were found to be downregulated amongst the WP users as compared to the control (NS) group.

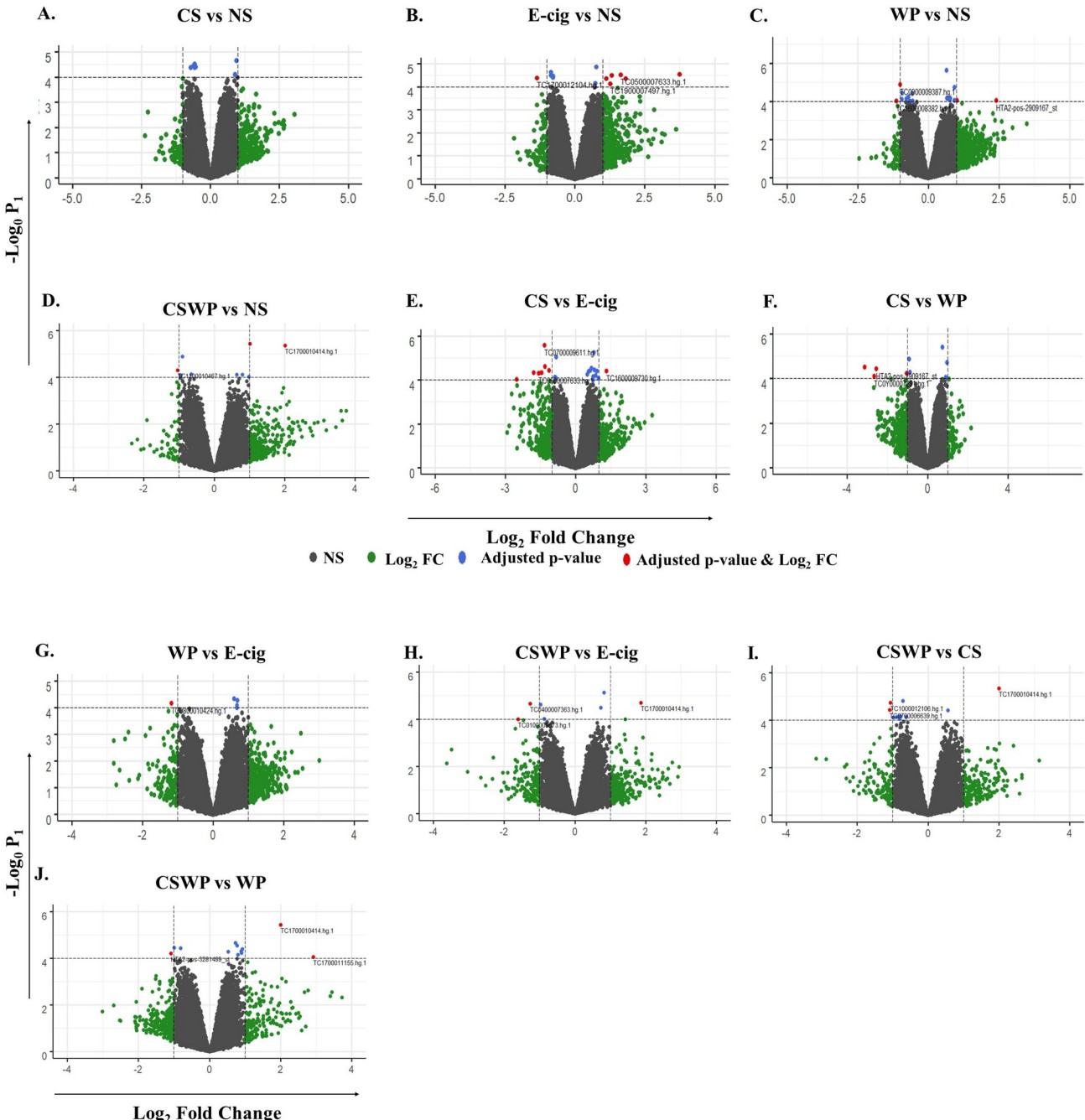

**Fig 3. Volcano plots showing number and distribution of lncRNAs.** Volcano plot showing the relation between p-values (Y-axis) vs fold change (X-axis) in the differentially expressed lncRNAs amongst (A) Cigarette smokers vs. non-smokers, (B) E-cig users vs. Non-smokers, (C) waterpipe smokers vs. Non-smokers, (D) dual smokers vs. Non-smokers, (E) Cigarette smokers vs. E-cig users, (F) Cigarette smokers vs. waterpipe smokers, (G) Waterpipe smokers vs. E-cig users, (H) Dual smokers vs. E-cig users, (I) Dual smokers vs. cigarette smokers, and (J) Dual smokers vs. waterpipe smokers. Log$_2$ fold change (cut-off = ± 1, vertical lines) was plotted against the–log$_{10}$ p-value (cut-off = 4, horizontal line). Adjusted p-value, $P <$ 0.0001.

**Dual smoker vs non-smokers.** On comparing lncRNA expression profiles of CSWP and NS groups, we found 9 differentially expressed lncRNAs in plasma exosomes, where four were downregulated and five upregulated in the plasma exosomes from dual smokers as compared

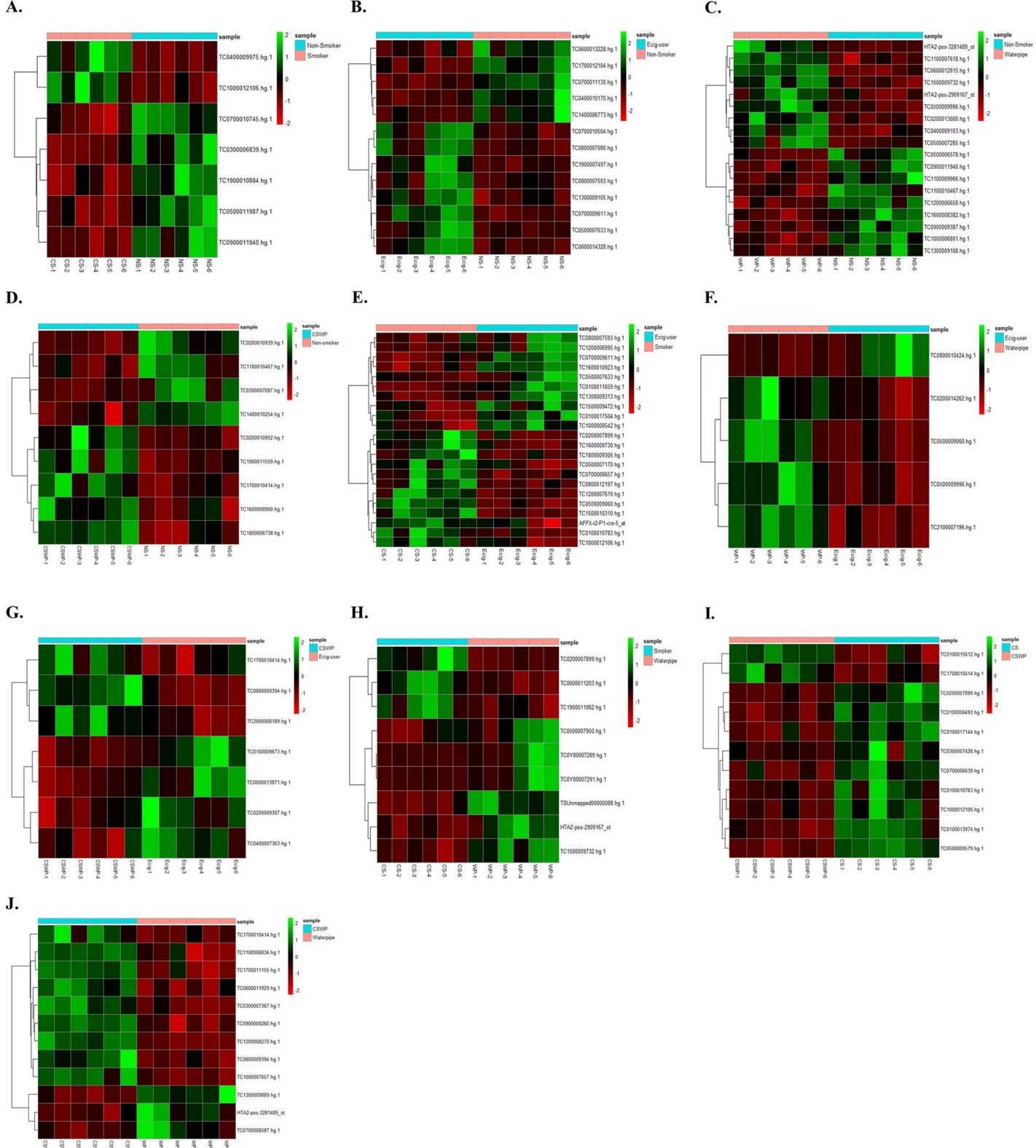

**Fig 4. Hierarchical cluster analyses of differentially expressed lncRNAs.** Heat map showing differentially expressed lncRNAs that are significantly varied between (A) Non-smokers and cigarette smokers, (B) Non-smokers and E-cigarette users, (C) Non-smokers and waterpipe users, (D) Non-smoker and dual smokers, (E) Cigarette

smokers and E-cig users, (F) E-cig users and waterpipe smokers, (G) Dual smokers and E-cig users, (H) Cigarette smokers and waterpipe smokers, (I) Cigarette smokers and dual smokers, and (J) Dual smokers and waterpipe users. These lncRNAs were identified based on individual pairwise comparisons (with unadjusted raw $p$-value; $P < 0.05$). The analysis was generated using Z scores of the most differentially expressed significant lncRNAs. Each row represents individual lncRNA, and each column represents individual sample. The relative lncRNA expression is depicted according to the color scale as shown on the right side of the figure. The magnitude of deviation from the median is represented by the color saturation. NS, non-smoker; CS, cigarette smoker; E-cig, Electronic cigarette users; WP, waterpipe smoker; CSWP: Dual Smoker.

to non-smoking controls (**Fig 4D**). Most of the differentially expressed lncRNAs identified on this comparison belonged to the family of putative proteins and thus have no known function.

**Smokers vs. E-cig users.** A total of 22 lncRNAs were found to show significant differential expression in E-cig users in comparison to CS group (**Fig 4E**). Of these, 10 lncRNAs were downregulated whereas 12 showed increased expression in cigarette smokers when compared to E-cig users. Many of the identified differentially expressed RNAs belonged to intronic or non-protein coding regions as shown in **Table 1**.

**Waterpipe vs E-cig users.** We demonstrate 5 differentially expressed lncRNAs (4 upregulated while 1 downregulated) in the plasma exosomes from WP in comparison to E-cig users as depicted in **Fig 4F** and **Table 1**.

**Dual smoker vs E-cig users.** We identified 4 downregulated and 3 upregulated lncRNAs, on comparing the lncRNA expression in plasma exosomes from dual smokers with those of E-cig users (**Fig 4G**).

**Smokers vs waterpipe users.** Nine lncRNAs were found to be differentially expressed in plasma exosomes from cigarette smokers as compared to WP group. Of these, most (6) were downregulated (**Fig 4H**). Importantly, two exon coding genes- rarsybo (Fold change = -3.14; p = 3.00E-05) and warbo (Fold change = -2.67, p = 7.88E-05)—showed three-fold decrease amongst CS group as compared to WP users.

**Dual smoker vs smokers.** We observed 11 differentially expressed lncRNAs on comparing the lncRNA expression between dual and cigarette smokers. Nine out of eleven of the differentially expressed lncRNAs were downregulated in dual smokers as compared to CS control (**Fig 4I**).

**Dual smoker vs waterpipe users.** Lastly, on comparing the lncRNA expressions between CSWP and WP groups we identified 12 differentially expressed lncRNAs. Most (9 out of 12) of the identified lncRNAs were upregulated in the plasma exosomes from dual smokers as compared to waterpipe users (**Fig 4J**).

## lncRNA expression overlaps amongst various treatment groups

To identify the lncRNAs that were commonly dysregulated on exposure to different types of exposures in our experiment, we performed a quick analysis of the differentially expressed lncRNAs and plotted a Venn diagrams to identify the common targets (**Fig 5A–5E**). The differentially expressed lncRNAs identified on comparing the expression profiles of the NS with E-cig users were distinct and did not show any overlap with other groups. One of the lncRNA gene ID (TC0900011940.hg.1) was found to be downregulated amongst both CS (Fold change = -0.72) and WP (Fold change = -0.71) groups when compared to NS. Similarly, another lncRNA (TC1100010467.hg.1) encoding for WT1 transcription factor was found to be downregulated amongst both waterpipe users (Fold change = -0.94) and dual smokers (Fold change = -1.04) when compared to NS (**Fig 5A**).

On plotting the results from comparing CS with rest of the study groups, we found few overlapping lncRNAs as shown in **Fig 5B**. Interestingly, the gene ID, TC0100010783.hg.1, was found to be commonly dysregulated in CS group when comparing with both E-cig and CSWP users. This gene encodes for the lnc-TDRD5, which has been shown to be responsible for chromatoid body assembly, retrotransposon silencing and spermiogenesis in mouse model [29].

**Table 1. Significant differentially expressed lncRNAs in various treatment groups and their significance.**

| PROBEID | GENENAME | log2 Fold change | P. Value | Description |
|---|---|---|---|---|
| **CS vs NS** | | | | |
| TC0300006839.hg.1 | uc062hmo.1 // AC020626.1 // — // — // — | -0.576946781 | 4.22E-05 | Archived gene ID |
| TC0400009975.hg.1 | poyvobo | 0.938840871 | 2.22E-05 | Homo sapiens single exon coding gene poyvobo |
| TC0500011987.hg.1 | AC004777.1 | -0.544358364 | 3.95E-05 | TEC tyrosine kinase |
| TC0700010745.hg.1 | AC007327.1 | -0.568070439 | 2.95E-05 | pseudogene similar to NADH dehydrogenase 2 MT-ND2 |
| TC0900011940.hg.1 | T368770 (miTranscriptome) | -0.719952578 | 4.04E-05 | Unannotated |
| TC1000012106.hg.1 | NA | 0.890647162 | 7.69E-05 | Unannotated |
| TC1900010884.hg.1 | NA | -0.606841977 | 3.49E-05 | Unannotated |
| **E-cig vs NS** | | | | |
| TC0400010170.hg.1 | skusworbu | -0.854161017 | 3.15E-05 | Putative Protein |
| TC0500007633.hg.1 | lozorby | 1.644179344 | 2.96E-05 | RNA-binding protein like family member. |
| TC0600013228.hg.1 | SLC2A12 | -0.792272819 | 3.95E-05 | Solute carrier family 2 member 12 |
| TC0600014328.hg.1 | OSTM1 | 0.764009224 | 1.32E-05 | Osteoclastogenesis associated transmembrane protein 1 |
| TC0700009611.hg.1 | Transfer RNA-Cys (GCA) 1–1 | 1.115171958 | 4.41E-05 | tRNA; Genhancer regulatory region |
| TC0700010504.hg.1 | OSBPL3 | 0.721080223 | 6.85E-05 | Oxysterol binding protein like 3 |
| TC0700011130.hg.1 | T324413 (miTranscriptome) | -0.862064073 | 2.30E-05 | Unannotated |
| TC0800007080.hg.1 | BNIP3L | 3.736036439 | 2.80E-05 | Bcl2 interacting protein 3 like |
| TC0800007593.hg.1 | zoyberby | 1.81372327 | 4.40E-05 | Endonuclease reverse transcriptase family member |
| TC1300009105.hg.1 | lnc-PCDH20-13 | 1.325335976 | 3.19E-05 | RNA gene |
| TC1400006773.hg.1 | miR4307 | -0.779299012 | 3.36E-05 | micro RNA |
| TC1700012104.hg.1 | T156883 (miTranscriptome) | -1.361512953 | 4.10E-05 | Unannotated |
| TC1900007497.hg.1 | flarchoy | 1.258862445 | 7.32E-05 | Endonuclease reverse transcriptase |
| **WP vs NS** | | | | |
| HTA2-pos-2909167_st | NA | 2.404401909 | 8.64E-05 | Normgene |
| HTA2-pos-3281489_st | NA | 1.012476113 | 8.85E-05 | normgene |
| TC0200013000.hg.1 | T191448 (miTranscriptome) | 0.760037466 | 8.17E-05 | Unannotated |
| TC0400009103.hg.1 | stoyswarby | 0.733276161 | 7.61E-05 | Spliced non-coding gene |
| TC0500006578.hg.1 | forgoy | -0.925005367 | 3.65E-05 | Putative Protein |
| TC0500007285.hg.1 | lnc-ZNF131-1 | 0.921228322 | 8.73E-05 | Antisense To ANXA2R |
| TC0600012815.hg.1 | PPIL6-201 | 0.639890503 | 2.29E-06 | Peptidylprolyl isomerase like 6 |
| TC0900009387.hg.1 | T354858 (miTranscriptome) | -1.006431605 | 1.31E-05 | Unannotated |
| TC0900011940.hg.1 | T368770 (miTranscriptome) | -0.710249664 | 5.84E-05 | Unannotated |
| TC0X00009996.hg.1 | AL590762.11 | 0.659766654 | 6.96E-05 | Pseudogene Similar To Part Of Poly(A) Binding Protein, Nuclear 1 (PABPN1) |
| TC1000006801.hg.1 | T037261 (miTranscriptome) | -0.777471484 | 7.50E-05 | Unannotated |
| TC1100007618.hg.1 | LOC100129915 | 0.780128991 | 6.59E-05 | Similar to ring finger protein 18 |
| TC1100009966.hg.1 | AC068733.2–201 | -0.576623305 | 3.90E-05 | TBC1 domain family, member 12, pseudogene |
| TC1100010467.hg.1 | WT1 | -0.941823433 | 8.02E-05 | WT1 transcription factor |
| TC1200006658.hg.1 | woybleyby | -0.658889026 | 9.38E-05 | Putative mitochondrial protein |
| TC1300009108.hg.1 | T095410 (miTranscriptome) | -0.55850742 | 9.48E-05 | Unannotated |
| TC1500009732.hg.1 | fawkloy | 0.920293174 | 1.79E-05 | Putative Protein |
| TC1600008382.hg.1 | T136042 (miTranscriptome) | -1.137693624 | 9.45E-05 | Unannotated |
| **CSWP vs NS** | | | | |
| TC0200010939.hg.1 | steeklu | -0.900124606 | 1.27E-05 | Putative protein |
| TC0200010952.hg.1 | NYAP2 | 0.63991652 | 7.66E-05 | Neuronal tyrosine-phosphorylated phosphoinositide-3-kinase adaptor 2 |
| TC0300007087.hg.1 | blerfarbo | -0.948070038 | 7.95E-05 | Putative protein |

*(Continued)*

**Table 1.** (Continued)

| PROBEID | GENENAME | log2 Fold change | P. Value | Description |
|---|---|---|---|---|
| **CS vs NS** | | | | |
| TC1100010467.hg.1 | WT1 | -1.042036815 | 4.89E-05 | WT1 transcription factor |
| TC1400010254.hg.1 | florpaw | -0.65229696 | 7.53E-05 | Putative protein |
| TC1600008900.hg.1 | sleeboby | 0.979624294 | 9.72E-05 | Putative protein |
| TC1700010414.hg.1 | starsmo | 2.016554814 | 4.44E-06 | Putative protein |
| TC1800006738.hg.1 | smeypla | 0.802532832 | 7.64E-05 | Putative protein of ancient origin |
| TC1900011559.hg.1 | T181991 (miTranscriptome) | 1.008317384 | 3.74E-06 | Unannotated |
| **CS vs E-cig** | | | | |
| AFFX-r2-P1-cre-5_at | NA | 0.559000778 | 4.04E-05 | Unannotated |
| TC0100010783.hg.1 | lnc-TDRD5 | 0.688322216 | 2.81E-05 | RNA gene |
| TC0100011659.hg.1 | lnc-MARK1-1 | -0.830260656 | 8.58E-06 | Non-protein coding lnc-MARK1-1:1 sequence. |
| TC0100017504.hg.1 | T031766 (miTranscriptome) | -0.846997861 | 7.80E-05 | Unannotated |
| TC0200007899.hg.1 | T191106 (miTranscriptome) | 0.785909826 | 5.65E-06 | Unannotated |
| TC0500007170.hg.1 | T281110 (miTranscriptome) | 0.745643686 | 6.86E-05 | Unannotated |
| TC0500007633.hg.1 | lozorby | -1.569509552 | 4.82E-05 | RNA binding protein like family member |
| TC0500009060.hg.1 | lnc-PCYOX1L-3 | 0.630448229 | 3.69E-05 | RNA gene |
| TC0700006657.hg.1 | Asparagine-linked glycosylation 1-like 5 | 0.806480376 | 3.48E-05 | Genhancer regulatory region; pseudogene |
| TC0700009611.hg.1 | Transfer RNA-Cys (GCA) 1–1 | -1.332318255 | 2.61E-06 | tRNA; Genhancer regulatory region |
| TC0800007593.hg.1 | zoyberby | -1.791751511 | 4.46E-05 | Endonuclease reverse transcriptase family member |
| TC0800012197.hg.1 | skorslubu | 0.910130832 | 3.89E-05 | Tigger transposable element-derived protein 1 like family member |
| TC1000008542.hg.1 | T046708 (miTranscriptome) | -1.137097908 | 3.62E-05 | Unannotated |
| TC1000012106.hg.1 | T050226 (miTranscriptome) | 0.98387836 | 8.45E-05 | Unannotated |
| TC1200006995.hg.1 | pleyblybu | -2.506882323 | 9.21E-05 | Spliced non-coding gene |
| TC1200007619.hg.1 | shanaw | 0.853525764 | 6.10E-05 | Putative protein |
| TC1300009313.hg.1 | seychoy | -1.443895461 | 4.70E-05 | Single exon coding gene |
| TC1500009472.hg.1 | NA | -0.880016936 | 7.51E-05 | Unannotated |
| TC1500010310.hg.1 | RNA, U7 small nuclear 79 pseudogene | 0.504790762 | 5.35E-05 | Genhancer regulatory region |
| TC1600009730.hg.1 | T128938 (miTranscriptome) | 1.326826001 | 3.77E-05 | Unannotated |
| TC1600010923.hg.1 | AC079414.1 | -1.309502335 | 2.38E-05 | Novel transcript, antisense to CLEC3A |
| TC1800009305.hg.1 | NONHSAG024018 92; Lnc-PIGN-6 | 0.748542125 | 9.87E-05 | Sense intronic non-coding RNA |
| **WP vs E-cig** | | | | |
| TC0200014262.hg.1 | T199176 (miTranscriptome) | 0.69106882 | 5.31E-05 | Unannotated |
| TC0500009060.hg.1 | lnc-PCYOX1L-3 | 0.590631549 | 4.53E-05 | RNA gene |
| TC0800010424.hg.1 | T343804 (miTranscriptome) | -1.184148902 | 6.68E-05 | Unannotated |
| TC0X00009996.hg.1 | AL590762.1 | 0.679813184 | 8.22E-05 | Pseudogene similar to part of poly(A) binding protein, nuclear 1 (PABPN1) |
| TC2100007196.hg.1 | T226511 (miTranscriptome) | 0.673584366 | 9.84E-05 | Unannotated |
| **CSWP vs E-cig** | | | | |
| TC0100009673.hg.1 | T018635 (miTranscriptome) | -1.604525926 | 9.79E-05 | Unannotated |
| TC0200009307.hg.1 | PRSS40A | -0.972747948 | 2.43E-05 | Serine protease 40A; pseudogene |
| TC0400007363.hg.1 | sheyflarby | -1.270572291 | 2.14E-05 | Putative protein |
| TC0600009394.hg.1 | bawplerbu | 0.816447817 | 7.32E-06 | Putative protein |
| TC0600013871.hg.1 | T315187 (miTranscriptome) | -0.871913931 | 9.62E-05 | Unannotated |
| TC1700010414.hg.1 | starsmo | 1.864419934 | 2.02E-05 | Putative protein |
| TC2000008189.hg.1 | T212697 (miTranscriptome) | 0.726204194 | 3.26E-05 | Unannotated |

*(Continued)*

**Table 1.** (Continued)

| PROBEID | GENENAME | log2 Fold change | P. Value | Description |
|---|---|---|---|---|
| **CS vs NS** | | | | |
| **CS vs WP** | | | | |
| HTA2-pos-2909167_st | NA | -2.559642238 | 3.56E-05 | normgene |
| TC0200007899.hg.1 | T191106 (miTranscriptome) | 0.730657071 | 3.84E-06 | Unannotated |
| TC0500007950.hg.1 | lnc-ATP6AP1L-5 | -1.050567649 | 5.72E-05 | RNA Gene; Genehancer regulatory elements. |
| TC0600011203.hg.1 | AL021918.1 | 0.958745545 | 1.92E-05 | WD repeat domain 59 (WDR59) pseudogene |
| TC0Y00007289.hg.1 | rarsybo | -3.144078758 | 3.00E-05 | Single exon coding gene. |
| TC0Y00007291.hg.1 | warbo | -2.666565528 | 7.88E-05 | Single exon coding gene. |
| TC1500009732.hg.1 | fawkloy | -0.934211892 | 1.35E-05 | Transcript Identified by AceView |
| TC1900011062.hg.1 | T178086 (miTranscriptome) | 0.892397234 | 8.58E-05 | Unannotated |
| TSUnmapped00000088.hg.1 | DUSP16 | -0.879332008 | 5.01E-05 | Dual specificity phosphatase 16 |
| **CSWP vs CS** | | | | |
| TC0100006493.hg.1 | smorarbo | -0.986103904 | 7.76E-05 | Single exon coding gene |
| TC0100010783.hg.1 | lnc-TDRD5 | -0.619045364 | 7.43E-05 | RNA gene |
| TC0100013974.hg.1 | RNF220 | -0.799985122 | 8.80E-05 | Ring finger protein 220 |
| TC0100015612.hg.1 | NONHSAT068475 | 0.554656506 | 3.89E-05 | Taqman probe, non-coding |
| TC0100017144.hg.1 | T029497 (miTranscriptome) | -0.895077856 | 7.51E-05 | Unannotated |
| TC0200007899.hg.1 | T191106 (miTranscriptome) | -0.710135209 | 1.57E-05 | Unannotated |
| TC0300007428.hg.1 | T242983 (miTranscriptome) | -0.783521911 | 6.85E-05 | Unannotated |
| TC0500009579.hg.1 | T294035 (miTranscriptome) | -0.820042957 | 7.02E-05 | Unannotated |
| TC0700006639.hg.1 | AC004895.1–203 | -1.083546595 | 3.71E-05 | Novel transcript |
| TC1000012106.hg.1 | T050226 (miTranscriptome) | -1.063392127 | 1.84E-05 | Unannotated |
| TC1700010414.hg.1 | starsmo | 2.004901352 | 4.52E-06 | Putative protein |
| **CSWP vs WP** | | | | |
| HTA2-pos-3281489_st | NA | -1.090364457 | 6.09257E-05 | Normgene |
| TC0300007367.hg.1 | KLHDC8B | 0.883254947 | 4.89501E-05 | Kelch domain containing 8B |
| TC0600009394.hg.1 | bawplerbu | 0.728763959 | 2.18613E-05 | Putative Protein |
| TC0600011929.hg.1 | lnc-CLIC5-2 | 0.522529849 | 5.27087E-05 | Genehancer region |
| TC0700008587.hg.1 | T330053 (miTranscriptome) | -0.994276685 | 3.44684E-05 | Unannotated |
| TC0900008260.hg.1 | T363928 (miTranscriptome) | 0.919987054 | 3.95504E-05 | Unannotated |
| TC1000007657.hg.1 | T042202 (miTranscriptome) | 0.803560023 | 7.08706E-05 | Unannotated |
| TC1100006834.hg.1 | vaslorby | 0.899127136 | 5.97471E-05 | Putative Protein |
| TC1200008270.hg.1 | lnc-NAV3-2:4 | 0.771483311 | 2.77469E-05 | Promoters and Genehancers |
| TC1300009889.hg.1 | T099623 (miTranscriptome) | -0.810946716 | 3.70585E-05 | Unannotated |
| TC1700010414.hg.1 | starsmo | 2.001800176 | 3.72921E-06 | Putative Protein |
| TC1700011155.hg.1 | NA | 2.923514656 | 8.83794E-05 | Unannotated |

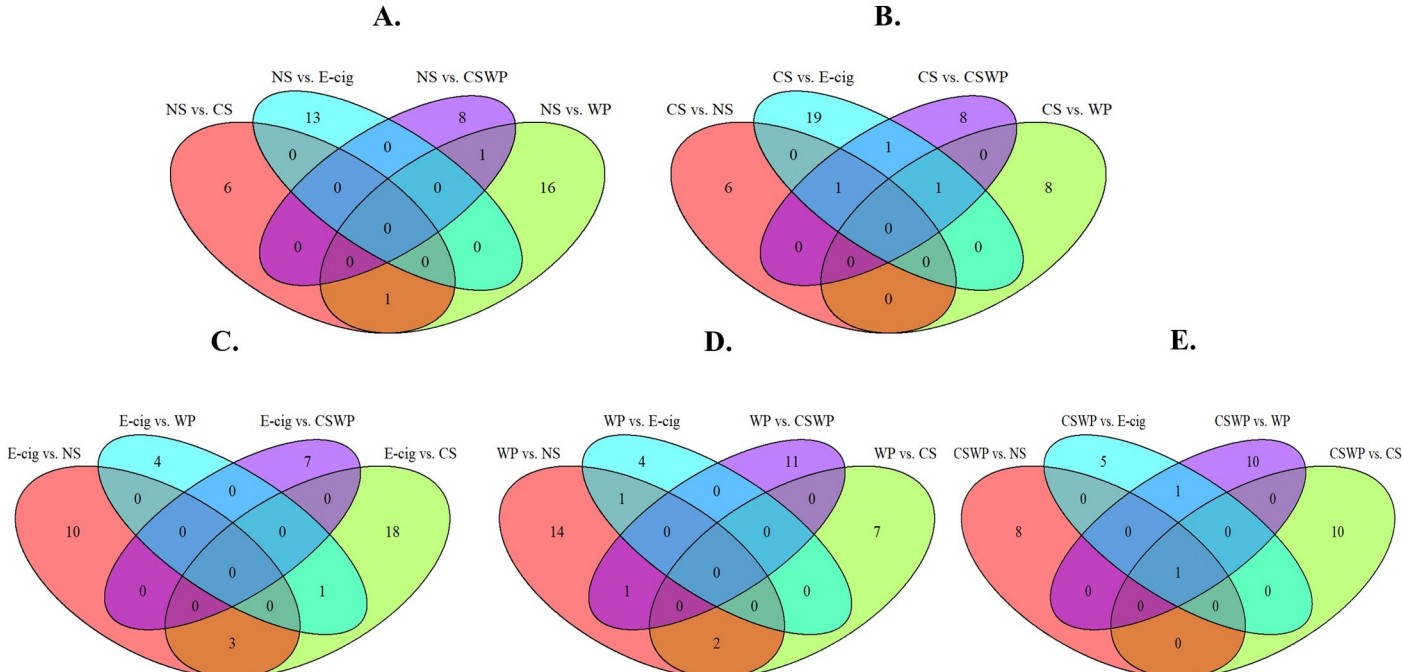

**Fig 5. Venn diagram showing the overlap of differentially expressed lncRNAs.** The overlap of differentially expressed lncRNAs between (A) non-smokers vs. cigarette smokers, E-cig users, waterpipe smokers or dual smokers. (B) cigarette smokers vs. non-smokers, E-cigarette users, waterpipe smokers or dual smokers, (C) E-cigarette users versus non-smokers, cigarette smokers, waterpipe smokers or dual smokers, (D) waterpipe smokers versus non-smokers, cigarette smokers, E-cigarette users or dual smokers, and (E) dual smokers vs. non-smokers, E-cig users, cigarette smokers or waterpipe smokers. NS, non-smoker; CS, cigarette smoker; E-cig, Electronic cigarette users; WP, waterpipe smoker; CSWP: Dual Smoker.

Similarly, when we compared the differentially regulated genes amongst E-cig users to other groups (NS, CS, WP and CSWP) we found three lncRNAs that were commonly dysregulated. These genes IDs were TC0500007633.hg.1, TC0700009611.hg.1, TC0800007593.hg.1 which encoded for RNA binding protein (lozorby), tRNA-Cys (GCA)1-1, and endonuclease reverse transcriptase (zoyberby) respectively (**Fig 5C**).

Comparisons made between WP users with other groups has revealed some common targets across various comparisons. For example, two of the genes like HTA2-pos-2909167_st and TC1500009732.hg.1 (identified as fawkloy) were common and up regulated in WP on comparison with NS and CS (**Fig 5D**).

While comparing the lncRNA expression profiles for dual smokers (CSWP) versus all the other exposure groups, we found a common gene (TC1700010414.hg.1) that was upregulated amongst dual smokers. This gene codes for a putative protein, starsmo, and has exhibited approximately 2-fold increase in CSWP group when compared to NS, CS, WP or E-cig users (**Fig 5E**).

## Functional analysis of differentially expressed lncRNAs in plasma exosomes from non-smokers, smokers, E-cig users, waterpipe users and dual smokers

Lastly, we performed FunRich gene enrichment analysis to perform functional annotation of the differentially expressed lncRNAs in various experimental groups. We found significant changes in the expression of lncRNAs involved in biological processes like, steroid metabolism, hemopoeisis and regulation of cell proliferation. The biological processes involving

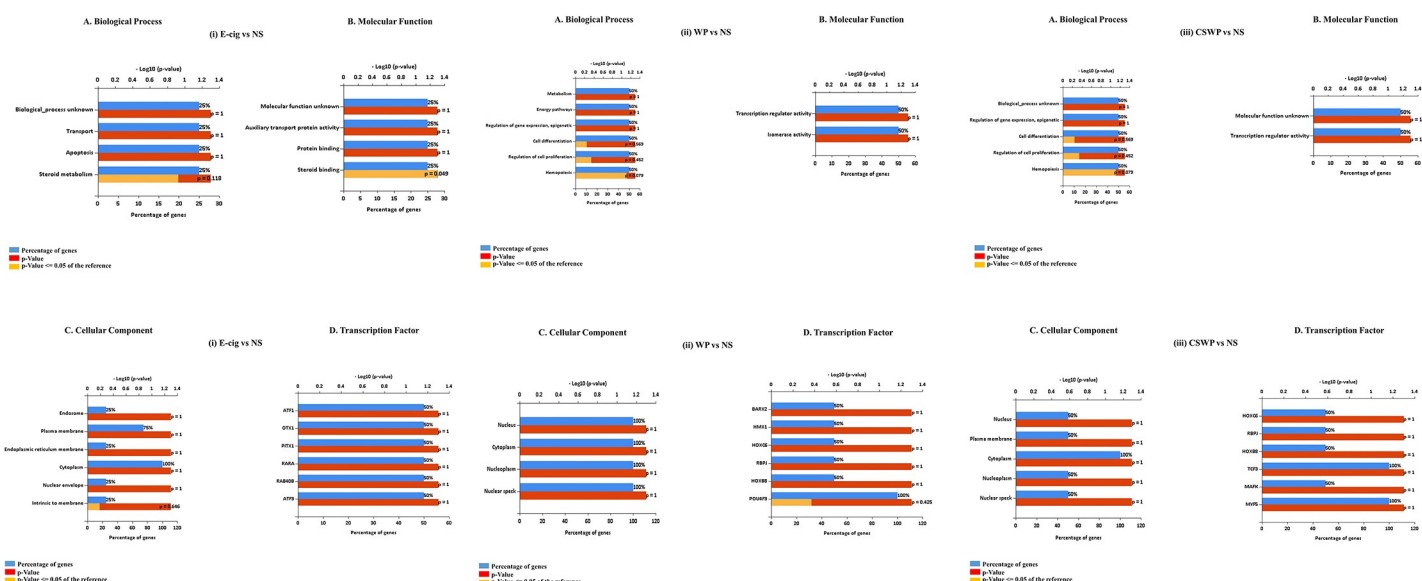

**Fig 6. FunRich gene enrichment analysis of the differentially expressed lncRNAs.** The top six enriched: (A) Biological process, (B) Molecular function, (C) Cellular component, and (D) Transcription factors for the significant lncRNAs and possible gene targets on pairwise comparisons between (i) E-cigarette users vs. Non-smokers, (ii) Waterpipe smokers vs. Non-smokers, and (iii) Dual smoker vs Non-smokers. NS, non-smoker; CS, cigarette smoker; E-cig, Electronic cigarette users; WP, waterpipe smoker; CSWP: Dual Smoker.

steroid metabolism are dysregulated amongst E-cig users as compared to non-smokers. When we enriched the differentially expressed genes based on their molecular functions, we again found a significant enrichment of genes involved in steroid binding which is enriched among the samples from E-cig users as compared to NS (**Fig 6A & 6B**).

Biological processes involved in cell differentiation and proliferation are significantly dysregulated amongst both waterpipe users and dual smokers as compared to non-smokers (**Fig 6A**). Most of the significant differentially expressed lncRNAs amongst E-cig users were intrinsic to membrane thus raising the possibility that E-cig use affects the cell membrane and related processes to the most (**Fig 6C**).

We did not find any significant change in the lncRNA expression when enriching the genes based on their involvement in various biological pathways or their site of expression (**S1 Fig**).

## Discussion

Exposure to cigarette smoke is associated with various chronic diseases such as atherosclerosis, COPD, and lung cancer [30–34]. Similarly, exposure to waterpipe smoke has also been strongly linked with complications like heart disease, chronic bronchitis, and cancers [35, 36]. Recently, there has been a rapid rise in the number of E-cig users especially among the youth. Despite being marketed or sold as a safer alternative to traditional tobacco smoking, recent reports have associated the severe acute pulmonary illnesses with symptoms of eosinophilic, hypersensitivity pneumonitis and acute lung injury with vaping [37–41]. Further, there are reports of oxidative stress, endothelial cell dysfunction [42, 43], and altered innate immune response in lungs of E-cig users [44]. In fact, we have recently reported the altered expression of systemic biomarkers of inflammation and endothelial dysfunction in E-cig users [26].

Based on our recent studies on profiling of miRNA in smokers, E-cig users, Waterpipe smokers and dual users [23], we compared the exosomal lncRNA profiles in the blood plasma from NS, CS, WP, CSWP and E-cig users. Genome-wide association studies have identified

distinct loci associated with pulmonary pathologies like COPD, asthma, and IPF [1, 11–14] These noncoding RNA (ncRNA) loci, encoding miRNAs, and lncRNAs, have been shown to play important roles in various cellular functions and diseases [1, 45–50]. Previous work by our lab has shown altered expression of exosomal miRNAs amongst our smoking and vaping groups [23]. Identifying the distinct lncRNA profiles amongst these groups was the main objective of this study that is critical in understanding the cellular mechanism altered following the use of various smoking/vaping products. The outcome of this study will further help in recognizing the pathologic biomarkers indicative of potential susceptibilities by vaping/smoking amongst these individuals.

We chose to use plasma-derived exosomes for this study as they are the most widely studied and extensively characterized exosomes [6, 51]. Blood plasma is easy to collect as compared to other biofluids like BALF and the method of isolation of exosomes (with high purity and yield) from blood plasma is well characterized [6, 52]. Furthermore, plasma and serum-derived exosomes have the potential to develop as biomarkers for various disease condition [51, 53, 54]. We found alterations in a distinct set of lncRNA on exposure to E-cig vapor, cigarette smoke, waterpipe smoke and dual smoke, with slight overlaps. We showed altered expression of 13 lncRNA gene loci among E-cig users as compared to NS. Importantly, we observed a four-fold increase in the TC0800007080.hg.1 gene locus that encodes Bcl2 interacting protein 3 like-protein (BNIP3L). BNIP3L, also known as Nix, is a mitochondrial protein that is shown to promote airway epithelial cell injury on exposure to cigarette smoke [55]. In fact, increased expression of BNIP3L points towards aggravated mitophagy, which is a hallmark of pulmonary disease conditions including COPD and IPF [56, 57]. Furthermore, in vivo studies reveal that BNIP3L is the pro-apoptotic transcriptional target of p53. p53, also known as the guardian of the genome, is a tumor suppressive gene most commonly mutated in case of human cancers [58]. Thus, our results showing an attenuation in the BNIP3L expression on E-cig use raises the possibility of emergence of COPD, IPF or even cancers in these individuals. Gene enrichment analyses revealed a significant change (p = 0.049) in the expression of lncRNAs involved in steroid binding in the plasma-derived exosomes from E-cig users as compared to their non-smoking controls. This supports the findings from previous studies that show disruption in lipid metabolism on E-cig use [59–61].

On comparing the lncRNA profiles between WP users and NS, we found 18 altered gene loci. Amongst the known targets were PPIL6 and WT1. PPIL6 encodes peptidylprolyl isomerase like-6 protein that has been shown to be associated with respiratory immune responses in animals like lambs and swine [62, 63]. In fact, FK506-binding protein (FKBP)-a member of this class of proteins- has altered expression in asthma. Furthermore, it is also involved in the dysregulation of pro/anti-inflammatory genes and signaling proteins in COPD. However, its exact role remains largely unknown [64].

Similarly, the expression of Wilms tumor 1 (WT1), a marker of mesothelial cells, is known to be associated with fibrotic lungs in various cell types [65]. Evidence suggests that even partial loss of WT1 results in abrogation of pleural mesothelial cell phenotype and results in myofibroblast accumulation with upregulation in the expression of profibrotic markers- α-smooth muscle actin (α-SMA) and fibronectin [66, 67]. Our result showing decreased expression of WT1 in both waterpipe and dual smoker groups points towards the possible emergence of pulmonary fibrosis amongst these users. Overall, our study shows altered expression of lncRNAs in various exposure groups and identifies targets that point towards a possible risk of lung injury and/or related pathology amongst these individuals. Future studies designed to unravel the role of each of the identified lncRNAs will be crucial in identification and development of biomarkers for obstructive and/or restrictive lung pathologies.

Nevertheless, it is pertinent to mention, that our study had some limitations. First, though the GeneChip WT Pico assay is designed to be highly accurate, it may suffer from the technical issues inherent to microarrays including cross-and/or non-specific hybridization and limited detection range of individual probes [68]. As a result, some of the targets may not have been identified in this study. Second, a majority of the significantly altered lncRNAs identified in our study were not annotated with their associated functions unknown. Many others were defined as encoding putative proteins/peptides and their role in cellular signaling and gene regulation remain largely elusive. Yet, the growing interest in the area has unraveled important biomarkers and regulatory ncRNAs which are associated with tobacco smoke-related pathologies [69]. Third, analysis of exosomal lncRNAs has a major limitation of not being able to ascertain the cellular origins of the lncRNAs, which can be helpful in pathophysiology of a particular disease condition. Analysis of cellular lncRNAs from various blood or lung cells may help in providing a more specific biomarker signature for smoke-exposure associated diseases in the future.

Nonetheless, the present study is the first ever attempt to compare the exosomal lncRNA profiles of individuals using tobacco cigarettes, E-cig, waterpipe or both (cigarette and waterpipe). We identified some important targets that might play an important role in regulating the inflammatory or autophagic responses on long-term use of these products. Future work in this area is required to understand the functions of these lncRNAs and deducing their roles in relation to chronic lung pathologies caused by vaping and/or smoking.

## Supporting information

**S1 Fig. FunRich gene enrichment analysis of the differentially expressed lncRNAs.** The top 6 enriched: (A) Biological pathway and (B) Site of expression for the significant lncRNAs and possible gene targets on pairwise comparisons between (i) E-cigarette users vs. Non-smokers, (ii) Waterpipe smokers vs. Non-smokers, and (iii) Dual smoker vs Non-smokers. NS, non-smoker; CS, cigarette smoker; E-cig, Electronic cigarette users; WP, waterpipe smoker; CSWP: Dual Smoker.
(PDF)

## Acknowledgments

We thank Ms Janice Gerloff and Dr. Naushad Ahmad Khan (isolated exosomes and sent exosomal RNA for the analysis to Thermo Fisher) for initial recruitment of subjects and technical assistance. We also thank Dr. Isaac Sundar for helpful discussion and for providing scientific inputs.

## Author Contributions

**Conceptualization:** Irfan Rahman.

**Data curation:** Gagandeep Kaur.

**Formal analysis:** Dongmei Li.

**Funding acquisition:** Irfan Rahman.

**Investigation:** Gagandeep Kaur, Kameshwar Singh, Krishna P. Maremanda.

**Methodology:** Gagandeep Kaur, Dongmei Li, Hitendra S. Chand.

**Project administration:** Irfan Rahman.

**Resources:** Kameshwar Singh, Krishna P. Maremanda, Hitendra S. Chand.

**Software:** Dongmei Li.

**Supervision:** Irfan Rahman.

**Writing – original draft:** Gagandeep Kaur.

**Writing – review & editing:** Kameshwar Singh, Krishna P. Maremanda, Dongmei Li, Hitendra S. Chand, Irfan Rahman.

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
