## [Decision Letter · Decision Letter 0]

25 Sep 2020

PONE-D-20-27081

Differential plasma exosomal long non-coding RNAs expression profiles and their emerging role in E-cig users, cigarette, waterpipe, and dual smokers

PLOS ONE

Dear Dr. Rahman,

Thank you for submitting your manuscript to PLOS ONE. After careful consideration, we feel that it has merit but does not fully meet PLOS ONE’s publication criteria as it currently stands. Therefore, we invite you to submit a revised version of the manuscript that addresses the points raised during the review process.

Please carefully revise your manuscript addressing each and every comment of the reviewers.

We look forward to receiving your revised manuscript.

Kind regards,

M. Firoze Khan, Ph.D.

Academic Editor

PLOS ONE

3. We note that you are reporting an analysis of a microarray, next-generation sequencing, or deep sequencing data set. PLOS requires that authors comply with field-specific standards for preparation, recording, and deposition of data in repositories appropriate to their field. Please upload these data to a stable, public repository (such as ArrayExpress, Gene Expression Omnibus (GEO), DNA Data Bank of Japan (DDBJ), NCBI GenBank, NCBI Sequence Read Archive, or EMBL Nucleotide Sequence Database (ENA)). In your revised cover letter, please provide the relevant accession numbers that may be used to access these data. For a full list of recommended repositories, see http://journals.plos.org/plosone/s/data-availability#loc-omics or http://journals.plos.org/plosone/s/data-availability#loc-sequencing

4. In your Methods section, please provide additional information about the demographic details of the patients included in the study. Please ensure you have provided sufficient details to replicate the analyses such as:

a) the date range (month and year) during which you collected specimens,

b) a description of how participants were recruited to provide samples, and

c) eligibility criteria for being included in this part of the study.

5. Thank you for stating the following above the Acknowledgments Section of your manuscript:

'Funding: This work was supported in part by a National Institutes of Health (NIH) Grants, NIH 1R01HL135613, and the National Cancer Institute of the NIH and the Food and Drug Administration (FDA) Center for Tobacco Products under Award Number U54CA228110.'

'The funders had no role in study design, data collection and analysis, decision to publish, or preparation of the manuscript.'

6. Please amend your list of authors on the manuscript to ensure that each author is linked to an affiliation. Authors’ affiliations should reflect the institution where the work was done (if authors moved subsequently, you can also list the new affiliation stating “current affiliation:….” as necessary).

7. Please include a copy of Table 1 which you refer to in your text on pages 9 and 10.

<h3>** **</h3>

8. We note you have included a table to which you do not refer in the text of your manuscript. Please ensure that you refer to Table 2 in your text; if accepted, production will need this reference to link the reader to the Table.

**Comments to the Author**

1. Is the manuscript technically sound, and do the data support the conclusions?

Reviewer #1: Yes

Reviewer #2: Yes

Reviewer #3: Yes

2. Has the statistical analysis been performed appropriately and rigorously? 

Reviewer #1: Yes

Reviewer #2: Yes

Reviewer #3: Yes

3. Have the authors made all data underlying the findings in their manuscript fully available?

Reviewer #1: Yes

Reviewer #2: Yes

Reviewer #3: Yes

4. Is the manuscript presented in an intelligible fashion and written in standard English?

Reviewer #1: Yes

Reviewer #2: Yes

Reviewer #3: Yes

5. Review Comments to the Author

Reviewer #1: This is an interesting pilot study using a human patient cohort evaluating the effect of smoking (water pipe, cigarette as well as e-cigs) on the expression profile of LncRNAs secreted in exossomes. The paper finds a consistent pattern of alteration of LncRNAs associated with pathways that regulate cellular differentiation and proliferation providing a potential explanation for the effects of smoking in promoting carcinogenesis and fibrosis in distal organs not directly in contact with the products of smoking. The experiments are appropriate and show solid results supporting the major conclusions despite the relatively small cohort. As the only critique figures seem copied directly from the screen of each instrument and their quality could be improved if appropriate graphic software was used.

Reviewer #2: The present manuscript describes lncRNA profiles in the plasma-derived exosomes from smokers, vapers, waterpipe users and dual smokers. This is an exciting study aiming to identify biomarkers related to chronic lung diseases as results from these smoking habits.

Minor comments:

1. It will be informative to discuss the rationale to use plasma-derived exosomes vs exosomes from airway secretions such as sputum, BAL etc.

2. The pros and cons of using GeneChip vs. RNAseq for lncRNA profiling need to be discussed.

3. Medical implications of the findings in the context of chronic obstructive or restrictive lung diseases need to be discussed.

Reviewer #3: In this study, the investigators have compared different groups of human subjects, namely non-smokers, cigarette smokers, e-Cig smokers, waterpipe smokers, and cigarette + waterpipe smokers. The goal was to identify differentially expressed long noncoding RNAs in the exposomes in the plasma, with an aim to identify potential liquid biopsy biomarkers. The experimental as well as computational analyses techniques are state-of-the art and the manuscript data are well interpreted. However, attention is drawn to the following deficiencies:

1. Metadata: This is a human subjects study and must include metadata on the sampled individuals such as their smoking history and other information as contained in the study questionnaires.

2. FIGURES: The current version of the figures has poor resolution and need to be fixed.

3. DISCUSSION: a)There is a need to further discuss the identified lncRNAs in the context of health and disease. How these noncoding RNA species could be implicated.

b). The relationship between the lncRNA species and their genomic context needs further elaboration.

4. The manuscript is well written but sporadic errors in language may be fixed.

6. PLOS authors have the option to publish the peer review history of their article (what does this mean?). If published, this will include your full peer review and any attached files.

Reviewer #1: No

Reviewer #2: No

Reviewer #3: No

---

## [Author Response · Author response to Decision Letter 0]

9 Nov 2020

PONE-D-20-27081

Differential plasma exosomal long non-coding RNAs expression profiles and their emerging role in E-cig users, cigarette, waterpipe, and dual smokers

Response: We thank the reviewers for expressing great interest in our study and appreciating our work. We have revised the manuscript based on reviewers’ comments, and believe that our revised manuscript is significantly improved based on the constructive comments provided by the reviewers. 

We have formatted and edited the manuscript as per the journal’s guidelines.

Response: The manuscript has now been thoroughly edited to rectify any grammatical or typographical errors.

3. We note that you are reporting an analysis of a microarray, next-generation sequencing, or deep sequencing data set. PLOS requires that authors comply with field-specific standards for preparation, recording, and deposition of data in repositories appropriate to their field. Please upload these data to a stable, public repository (such as ArrayExpress, Gene Expression Omnibus (GEO), DNA Data Bank of Japan (DDBJ), NCBI GenBank, NCBI Sequence Read Archive, or EMBL Nucleotide Sequence Database (ENA)). In your revised cover letter, please provide the relevant accession numbers that may be used to access these data. For a full list of recommended repositories, see http://journals.plos.org/plosone/s/data-availability#loc-omics or http://journals.plos.org/plosone/s/data-availability#loc-sequencing

Response: We thank the reviewers for this suggestion. We have now deposited the data to GEO repository and the accession number is GSE160769. This information is now included in the manuscript. 

4. In your Methods section, please provide additional information about the demographic details of the patients included in the study. Please ensure you have provided sufficient details to replicate the analyses such as:

a) the date range (month and year) during which you collected specimens,

b) a description of how participants were recruited to provide samples, and

c) eligibility criteria for being included in this part of the study.

Response: We have now included the aforementioned information in the Methods section of the manuscript with proper referencing to our previous publication reporting a detailed description of the same.

5. Thank you for stating the following above the Acknowledgments Section of your manuscript:

'Funding: This work was supported in part by a National Institutes of Health (NIH) Grants, NIH 1R01HL135613, and the National Cancer Institute of the NIH and the Food and Drug Administration (FDA) Center for Tobacco Products under Award Number U54CA228110.'

'The funders had no role in study design, data collection and analysis, decision to publish, or preparation of the manuscript.'

Response: We have now removed the Funding information from the manuscript and updated the information under the Funding Statement in the online submission form, and in the cover letter.

Response: The required changes have now been included. 

The following is included in the cover letter as well as in the metadata online portal for your perusal:

Funding: This work was supported in part by a National Institutes of Health (NIH) Grants, NIH 1R01HL135613, and the National Cancer Institute of the NIH and the Food and Drug Administration (FDA) Center for Tobacco Products under Award Number U54CA228110.

The funders had no role in study design, data collection and analysis, decision to

publish, or preparation of the manuscript.

6. Please amend your list of authors on the manuscript to ensure that each author is linked to an affiliation. Authors’ affiliations should reflect the institution where the work was done (if authors moved subsequently, you can also list the new affiliation stating “current affiliation:….” as necessary).

Response: The Authors’ affiliation is updated in the manuscript. 

7. Please include a copy of Table 1 which you refer to in your text on pages 9 and 10.

Response: We have now included a copy of Table 1 within the manuscript.

8. We note you have included a table to which you do not refer in the text of your manuscript. Please ensure that you refer to Table 2 in your text; if accepted, production will need this reference to link the reader to the Table.

Response: We would like to confirm that we do not have a Table 2 in this manuscript. We have now rectified the typographical error in the manuscript. 

Comments to the Author

Reviewer #1

This is an interesting pilot study using a human patient cohort evaluating the effect of smoking (water pipe, cigarette as well as e-cigs) on the expression profile of lncRNAs secreted in exosomes. The paper finds a consistent pattern of alteration of lncRNAs associated with pathways that regulate cellular differentiation and proliferation providing a potential explanation for the effects of smoking in promoting carcinogenesis and fibrosis in distal organs not directly in contact with the products of smoking. The experiments are appropriate and show solid results supporting the major conclusions despite the relatively small cohort. As the only critique figures seem copied directly from the screen of each instrument and their quality could be improved if appropriate graphic software was used.

Response: We thank the reviewer for their feedback and suggestions. We have now edited the figure file and used original high-resolution figures for the PowerPoint slides.

Reviewer #2

The present manuscript describes lncRNA profiles in the plasma-derived exosomes from smokers, vapers, waterpipe users and dual smokers. This is an exciting study aiming to identify biomarkers related to chronic lung diseases as results from these smoking habits.

Minor Comments: 

1. It will be informative to discuss the rationale to use plasma-derived exosomes vs exosomes from airway secretions such as sputum, BAL etc.

Response: We thank the reviewer for this comment. The purpose of this study was to compare the lncRNAs profile amongst non-smokers, smokers, e-cig users, water pipe users and dual smokers for the purpose of identification of potential biomarkers. We chose to use plasma-derived exosomes as they are the most extensively characterized types of exosomes. The reason for using plasma-derived exosomes over other bio-fluids include; (a) ease of sample collection, (b) effective exosomal isolation, and (c) good yield. We have now included this information in the manuscript as well. 

2. The pros and cons of using GeneChip vs. RNAseq for lncRNAs profiling need to be discussed.

Response: We employed GeneChip™ WT Pico kit for this study. This is a highly sensitive and flexible assay that can be used to assess the RNA expression from as little as 100 picogram of total RNA. The assay ensures strand specificity and can perform multiple layers of analysis in combination with transcriptome arrays to accurately measure gene- or transcript-level expression of coding as well as long non-coding RNA. Though the assay is designed to be highly accurate, it may suffer from the technical issues inherent to microarrays including cross-and/or non-specific hybridization and limited detection range of individual probes. In spite of these shortcomings, we chose GeneChip over RNAseq as we were interested in using a quick, affordable, and user-friendly assay to compare the lncRNA profiles in our study groups. GeneChip with its user-friendly platform for data analyses enabled us to screen and analyze our samples quickly and efficiently. Since sequencing depths and finding newer transcripts was not the main purpose of this study, we did not choose to conduct RNAseq on our samples. We have now included our rationale in the manuscript as well and would like to thank the reviewer for this comment.

3. Medical implications of the findings in the context of chronic obstructive or restrictive lung diseases need to be discussed.

Response: We thank the reviewer for this comment. Our study identified targets like PPIL6, WT1 and BNIP3L amongst users of e-cigs, waterpipe or dual smokers. These targets have been known to be altered amongst patients with COPD, IPF and lung cancers. In fact, out study results are in conjunction with the existing literature and raise the possibility that these lncRNAs could serve as a biomarker for chronic obstructive or restrictive lung diseases amongst smokers, e-cig users or waterpipe users. The medical implications of our findings have now been clearly discussed in the manuscript.

Reviewer #3

In this study, the investigators have compared different groups of human subjects, namely non-smokers, cigarette smokers, e-Cig smokers, waterpipe smokers, and cigarette + waterpipe smokers. The goal was to identify differentially expressed long noncoding RNAs in the exosomes in the plasma, with an aim to identify potential liquid biopsy biomarkers. The experimental as well as computational analyses techniques are state-of-the art and the manuscript data are well interpreted. However, attention is drawn to the following deficiencies:

1. Metadata: This is a human subjects’ study and must include metadata on the sampled individuals such as their smoking history and other information as contained in the study questionnaires.

Response: We thank the reviewer for pointing this out. The study subject recruitment questionnaire and inclusion and exclusion criteria have been previously reported. However, we have now included a brief description about the subject recruitment criteria and referred to our previous publication to clearly define the subject inclusion/exclusion for this study.

2. FIGURES: The current version of the figures has poor resolution and need to be fixed. 

Response: We thank the reviewer for this observation. We have now changed the figure file and included original figures in the revised manuscript. 

3. DISCUSSION: 

a) There is a need to further discuss the identified lncRNAs in the context of health and disease. How these noncoding RNA species could be implicated.

Response: We thank the reviewer for pointing this out. We have now included the implications of our findings in terms of health and disease in the Discussion section of the manuscript. Our findings clearly indicate the increased susceptibility towards chronic obstructive and restrictive disease among smokers, e-cig users, waterpipe users and dual smokers and identifies the lncRNAs that could develop as biomarkers for the respective diseases in the future.

b). The relationship between the lncRNAs species and their genomic context needs further elaboration.

Response: We thank the reviewer for this comment. We have now included further information about the genomic context of each of the lncRNAs. 

4. The manuscript is well written but sporadic errors in language may be fixed.

Response: We thank the reviewer for identifying this. We have now thoroughly edited the manuscript to correct the sporadic language errors and other typographical mistakes.

---

## [Editor Report · Decision Letter 1]

16 Nov 2020

Differential plasma exosomal long non-coding RNAs expression profiles and their emerging role in E-cig users, cigarette, waterpipe, and dual smokers

PONE-D-20-27081R1

Dear Dr. Rahman,

We’re pleased to inform you that your manuscript has been judged scientifically suitable for publication and will be formally accepted for publication once it meets all outstanding technical requirements.

Kind regards,

M. Firoze Khan, Ph.D.

Academic Editor

PLOS ONE
---

## [Editor Report · Acceptance letter]

23 Nov 2020

PONE-D-20-27081R1 

 Differential plasma exosomal long non-coding RNAs expression profiles and their emerging role in E-cigarette users, cigarette, waterpipe, and dual smokers

Dear Dr. Rahman:

I'm pleased to inform you that your manuscript has been deemed suitable for publication in PLOS ONE. Congratulations! Your manuscript is now with our production department. 

Kind regards, 

on behalf of

Dr. M. Firoze Khan 

Academic Editor

PLOS ONE